# Do CEO Attributes Spur Conservatism?

Rawan Atwa [1,*], Safaa Alsmadi [1], Buthiena Kharabsheh [2] and Ruwaidah Haddad [1]

1 Faculty of Business, Accounting Department, Yarmouk University, P.O. Box 566, Irbid 21163, Jordan
2 Faculty of Business, Finance Department, Yarmouk University, P.O. Box 566, Irbid 21163, Jordan
* Correspondence: rawan@yu.edu.jo

**Abstract:** This study examines the relationship between chief executive officers' (CEOs') characteristics (e.g., tenure, experience, education, age and compensation) and accounting conservatism for a sample of 672 yearly observations from both Jordanian industrial and service companies listed on the Amman Stock Exchange (ASE) during the period 2014–2021. Using feasible generalised least squares, the results show that CEOs with more experience and skills are positively and significantly related to accounting conservatism. Furthermore, consistent with upper-echelon-theory arguments, the findings reveal that CEO tenure is significantly and positively associated with the level of accounting conservatism. The results indicate that CEOs' education, age and compensation are positively but insignificantly related to accounting conservatism. Overall, this study contributes to the literature by providing evidence of the importance of recognising the effects of CEOs' characteristics on influencing accounting conservatism in Jordanian industrial and service companies.

**Keywords:** CEO characteristics; accounting conservatism; Jordan; Amman Stock Exchange

## 1. Introduction

Valuable information about the firm's operating, investing and financial activities is usually in the hands of management. Furthermore, boards function on information provided by CEOs (Aram and Cowen 1983); more importantly, according to Bertrand and Schoar (2003) and Bamber et al. (2010), managers can have the discretion to affect and change corporate decisions to fulfil their own interests.

CEO-specific styles (i.e., norms, habits and values) are expected to affect corporate decisions (Bertrand and Schoar 2003; Malmendier and Tate 2005) and financial reporting decisions (Faulkner et al. 2020). While upper-echelon theory (Hambrick and Mason 1984) expects managers' personal attributes to affect corporate decisions, agency theory (Jensen and Meckling 1976) expects that their personal interests can come into play while conducting their responsibilities. Interests such as increasing compensation and enhancing one's personal reputation can lead managers to not report or delay reporting information if they expect it to negatively affect their interests. Managers can use accounting methods that result in an increase in their own income, thus affecting the quality of the information disclosed (Watts and Zimmerman 1986).

Efforts by standard setting bodies have been made to alleviate managers' attempts to manipulate information and ensure the relevant and fair representation of financial statements. According to Sterling (1970), conservatism is the most influential accounting standard of valuation. Despite the recent opposition to conservatism, Watts (2003) reported that empirical research suggests that accounting practices have become more conservative. Watts argues that the survival of conservatism proves its consequential benefits despite the arguments over its limitations.

In general, research into the effect of managerial characteristics on accounting conservatism is limited; the available research has focused on country- and firm-specific factors (Zhong and Li 2017). The characteristics of the management team are of great importance when analysing the determinants of accounting conservatism; for example, according to

Francis et al. (2015), managers' attitudes towards risk are reflected in accounting conservatism. However, research on the effect of managerial qualities on the level of accounting conservatism is limited, and the existing research has focused on countries with the largest economies, such as the US (Amin et al. 2022) and China (Yin et al. 2020). Moreover, qualities that have been examined include facial masculinity (Amin et al. 2022), religion (Ma et al. 2020), and the experience of famine (Hu et al. 2020). Further, characteristics such as age, tenure and education are still under-researched.

In Jordan, recent studies have reported an increase in conservative financial reporting practices (Alkordi et al. 2017; Makhlouf et al. 2018). Although other studies in Jordan have focused on the effect of the board on conservatism (Makhlouf et al. 2018) and ownership structure (Alkordi et al. 2017), to the best of the authors' knowledge, no research has focused on the effect of CEO qualities on conservatism in Jordan.

This research aims to fill this gap in the literature of managerial attributes and conservatism, examining the effect of CEO age, tenure, education, experience and compensation on the level of accounting conservatism in Jordan by employing a regression analysis and using income accruals as a proxy for conservatism. We find that CEO age and experience were positively associated with the level of conservatism. Therefore, the study adds to the limited research on the relation between CEO attributes and conservatism by proving that CEO age and tenure are important factors that affect conservatism.

Examining the relationship between CEO attributes and conservatism in Jordan is important for several reasons. First, it is consequential for market participants to understand how CEO characteristics can affect the quality of financial reporting for firms listed on the Amman Stock Exchange (ASE); ASE in 2022 had a market capitalisation of JOD 18 billion, with 48.1 percent of the listed shares owned by foreign investors (CEIC 2023). Second, studies in the Jordanian context can be of great importance to market players in other developing countries, specifically countries in the Middle East and the Arab region given that they share similar institutional and cultural factors. Therefore, the findings of the current research can help market players, not only in Jordan, make their investment decisions by providing results on how CEO attributes can affect the quality of financial reporting.

The contributions of this research are threefold. First, to the best of the authors' knowledge, it is the first study in the Arab world to address the effect of CEO characteristics. Therefore, it adds to the accounting literature on the effect of CEO attributes on conservatism in the Arab region. Second, this research extends the financial reporting quality literature by identifying CEO tenure and experience as determinants of financial reporting quality. Third, this research provides practical implications to policymakers, investors and owners in Jordanian corporations and the Jordanian market. Specifically, policymakers are advised to consider CEO characteristics when revising governance rules and regulations, while owners and directors are advised to focus on the experience of their managers rather than their education, which was found to be insignificant to the level of conservatism. Investors, before making their investment decisions, are recommended to consider the qualities of CEOs.

The remainder of the study continues as follows. In Section 2, we present the theoretical background, a review of the literature and the development of our hypotheses. Section 3 discusses the methodology employed. In Section 4, we present results and the discussion of the results. Finally, Section 5 concludes the research.

## 2. Literature Review and Development of Hypotheses

There is no universally accepted definition of *accounting conservatism*, mainly because it is still arguable whether conservatism is a useful financial reporting quality. On one hand, *accounting conservatism* is defined as the exercise of being cautious when recognising and measuring the earnings and assets of the firm (Watts 2003; Ball et al. 2013). Conservatism requires a higher degree of verification when recognising good news compared with bad news (Basu 1997). This reduces managerial bias in financial reporting, restricts opportunistic

payments, reduces managers' discretion to manipulate results (Dechow et al. 2010) and reduces litigation risk, which is more likely in the case of overstated net assets.

On the other hand, conservativism leads to downwardly biased results, which bring about lower cumulative retained earnings (Ahmed et al. 2002). Conservative financial reporting could underestimate the future value of an organisation through overexpensing or early expensing, deferring the recognition of revenue and lowering costs or the market values of inventory and asset impairments (Brown et al. 2006; Gigler et al. 2009). Moreover, it has been argued that stakeholders are interested in timely reporting on gains and losses alike, raising doubts about this one-sided application of conservatism (Gigler et al. 2009). In addition, Salehi and Azimi (2022) found that conservatism is not correlated with the level of information asymmetry; in other words, conservative accounting practices do not necessarily result in less private information. Despite criticism, conservatism has survived in accounting for many centuries and appears to have increased in the past 30 years.

Two arguments can be found on the tendency of CEOs to practice conservatism in their financial performance practices. CEOs tend to be conservative to signal their competency to the market. Simply put, CEOs, having better experience and competency, are expected not only to enhance firm performance but also to predict variation in future performance (Baik et al. 2011), anticipate future losses and incorporate them in a timely manner. Therefore, as a signal to the market, executive managers are expected to exercise higher levels of conservatism. On the other hand, because managers' interests conflict with those of the owners (Al-Maliki et al. 2022), managers might try to increase earnings by applying aggressive accounting practices that do not require strict verification for the recognition of good news to enhance their private benefits, especially if managerial compensation is linked to reported earnings, as argued by Basu (1997). Cyert and March (1963) hypothesise that complex and strategic decisions are not only based on economic optimisation but also, to a high extent, caused by behavioural factors. By the same token, upper-echelon theory suggests that top managements' values, experiences and mental processes affect their organisational outcomes (Hambrick and Mason 1984).

However, the literature on managerial attributes and organisational outcomes is limited. Law and Mills (2017) examined the effect of CEOs' military experience and tax avoidance, and Wowak et al. (2022) went on to examine the effect of CEO activism on the attitudes and behaviours of the employees of the firm.

A stream of research has examined the effect of CEO attributes on accounting conservatism, where researchers have focused on CEO gender in the US and China (Ho et al. 2015; Li et al. 2022), facial masculinity of CEOs in the US (Amin et al. 2022), managerial entrenchment in Iran (Salehi et al. 2021), CEO compensation in the US (Sikalidis 2021) and CEOs' social ties with their boards in China (Yin et al. 2020). To date, research on the relationship between CEO attributes and accounting conservatism has been, first, focused on developed countries; second, focused on attributes such as gender and facial masculinity; and third, focused on attributes related to a very specific situations, such as the great Chinese famine (Hu et al. 2020). The current research is an attempt to fill the gap by focusing on a developing country such as Jordan and by examining several attributes that have not been explored in the literature, namely age, tenure, education, experience and compensation.

### 2.1. CEO Age and Accounting Conservatism

Young managers are seen to be relatively greater risk takers and are more motivated to devise new and unprecedented ideas, in contrast with older managers, who are deemed to be conservative. Darmadi (2011) has found that young managers are less conservative and more open to novel insights. In the same vein, Horváth and Spirollari (2012) found that older managers prefer investments with quick returns and are less willing to take risks.

Upper-echelon theory (Hambrick and Mason 1984) provides three explanations as to why old managers are more likely to be conservative. First, older managers need more time to grasp new ideas and behaviours (Chown 1960). Growing older, the ability to integrate information into decisions becomes slower; however, the tendency to take more time and

information to evaluate and make decisions increases. Second, older managers are more committed to the status quo of the firm (Stevens et al. 1978). Finally, older managers are expected to avoid risky behaviours that might jeopardise their well-established careers and financial stability (Carlsson and Karlsson 1970).

Empirically, Bamber et al. (2010) and Degenhart et al. (2021) found that older managers are associated with more conservative accounting disclosures. While Yu (2021) found that CFO age is negatively related to the level of conservatism, Adyunita et al. (2021) did not find any significant association between a board members' age and their conservatism. In Jordan, Makhlouf et al. (2018) reported a positive yet insignificant relationship between the age of the board members and accounting conservatism.

On the basis of the above discussion, the following is expected:

**Hypothesis 1 (H$_1$).** *CEO age is significantly and positively associated with the level of conservatism.*

### 2.2. CEO Tenure and Accounting Conservatism

During the early years of a CEO's service, it is expected that the market is still uncertain about their skills and capabilities; accordingly, the results at that time are more likely to affect the market's perception of their competency (Fama 1980). Such a perception is of great significance to CEOs because it affects their future chances of appointment and their compensation. Therefore, at the beginning of their career, CEOs are expected to engage in opportunistic and aggressive behaviour in order to send signals to the marketplace regarding their competencies.

Moreover, executives will be forced out if they fail to achieve good results (Brown 1982; Pfeffer and Leblebici 1973), while CEOs stay for longer periods if they have achieved good results in the past. Such CEOs are expected to have an established reputation of experience to protect; therefore, the longer a CEO's tenure is at a firm, the more they have to lose. Moreover, tenure and employee commitment are positively associated (Buchanan 1974). In contrast, CEOs with less tenure, according to previous research, tend to make more changes in structure, procedures and people compared with long-serving CEOs (Carlson 1972; Helmich and Brown 1972; Kotin and Sharaf 1967); therefore, long-serving CEOs are more likely to be conservative and less prone to venturesomeness.

Upper-echelon theory (Hambrick and Mason 1984) suggests that the longer a CEO stays in their firm, the more likely their perspective and knowledge base will become limited and their options and strategies will be less innovative.

Literature on the relationship between CEO tenure and conservatism provided mixed results; Ali and Zhang (2015) found that CEOs in the early years of their service tend to overstate earnings, while Dong and Gai (2017) found that executive tenure, specifically that of the CFO, is positively correlated with the proportion of asset impairment provision. In contrast, Muttakin et al. (2019) reported a negative relation. Therefore, we expect that long-serving CEOs who are well situated in their firm and industry and in long working relationships will be conservative.

On the basis of the above discussion, the following is expected:

**Hypothesis 2 (H$_2$).** *CEO tenure is significantly and positively associated with the level of conservatism.*

### 2.3. CEO Education and Accounting Conservatism

Pfeffer (1981) argues that education in management serves only as a filtering device for staffing and enrolling individuals to jobs; in other words, such degrees will not have any consequential effect in the long-term for the individual or the firm.

According to upper-echelon theory, business schools focus on analytical techniques that are geared primarily towards avoiding big losses or mistakes; therefore, professional education in management is associated with moderation (Hambrick and Mason 1984). Moreover, the holders of such degrees are expected not be innovative or prone to risk compared with "self-made" executives (Collins and Moore 1970).

They argue that different functional tracks are expected to influence CEOs' strategic choices; for example, they assume that CEOs experienced in 'throughout functions', such as accounting, tend to focus on improving the efficiency of the process; this is compared to those experienced in 'output functions', such as marketing and sales, who are expected to focus on growth and finding new opportunities. If this holds true, CEOs with an accounting major are expected to keep a wary eye on the effectiveness and efficiency of operations, hence reducing the incentives from aggressive accounting practices.

The empirical literature has reported that CEOs with higher education levels show a higher level of accounting conservatism compared with their noneducated counterparts (Ason et al. 2021). It was found that financial expertise in the audit committee is positively correlated with conservatism (Krishnan and Visvanathan 2008). Makhlouf et al. (2018) found that the education level of the Jordanian board members is positively related to conservatism.

On the basis of the above discussion, the following is expected:

**Hypothesis 3 (H$_3$).** *CEO education is significantly and positively associated with the level of conservatism.*

### 2.4. CEO Experience and Accounting Conservatism

Experienced managers are expected to effectively perform complex tasks (Mishra 2014). For example, as Haider et al. (2021) argue, experienced managers are expected to efficiently deal with firms' contracts, such as debt contracts; therefore, they are likely to incorporate, in a timely manner, economic losses into financial statements. Moreover, experienced managers have more to lose in the case of failure to provide timely information to various users of the financial statements; such managers are risking their reputation in the market. Therefore, experienced managers are less likely to engage in opportunistic and aggressive activities or expropriate the firm's resources.

On the other hand, upper-echelon theory (Hambrick and Mason 1984) expects that the career experiences of executive management partially shape the lenses through which they view opportunities and problems and become part of their cognitive and emotional givens. It suggests that an executive whose experience is in one organisation is expected to have only relatively limited perspectives. This is in contrast with executives with experience from outside the firm, who are expected to be more innovative and to have a wider perspective. However, albeit innovative, experienced managers have future careers and reputations to care for.

In their review of 60 studies of executives' characteristics, Plöckinger et al. (2016) reported that experience, among other attributes, reduces risk tolerance in financial reporting. Haider et al. (2021) reported that managers with experience tend to report more conservatively; Yunos et al. (2014) found that experienced members on the board increase conservatism; and Chouaibi and Chiekh (2017) reported that CEO experience has an insignificant effect on conservatism.

On the basis of the above discussion, the following is expected:

**Hypothesis 4 (H$_4$).** *CEO experience is significantly and positively associated with the level of conservatism.*

### 2.5. CEO Managerial Compensation and Accounting Conservatism

Jensen and Meckling (1976) and Watts and Zimmerman (1990) found that managers are motivated to fudge their disclosures if those disclosures would affect their compensation. This paper distinguishes between two types of CEO compensations: basic salary and equity-based compensation.

We argue that managers who make higher salaries from a firm are more likely to be dependent on its success and continuity (Lewellyn et al. 1969; Lewellen and Huntsman 1970); furthermore, they jeopardise their job if the firm's performance deteriorates (Salancik and Pfeffer 1980). Simply put, we argue that the higher the salary of the manager, the more

valuable the firm's income becomes; in such a case, managers are less likely to run the risk of being fired for applying aggressive accounting. In fact, Desai et al. (2006) found that managers of firms involved with accounting restatements are significantly less likely to later be appointed to comparable positions.

With equity-based compensation, managers are expected to use aggressive accounting and to overvalue the firm for self-interest, to increase their compensation. It has been argued that managers engage in risky activities when they receive equity-based compensation (Jensen and Meckling 1976), which is consistent with Cohen et al.'s (2000) work, which reported that stock options motivate managers to take risk.

According to agency theory (Jensen and Meckling 1976), managers are expected to be opportunistic and to act in their self-interest. Therefore, in cases where manager compensation is equity based, it is expected that managers will be less conservative. On the other hand, upper-echelon theory (Hambrick and Mason 1984) expects that managers, owners or nonowners, are less likely to risk the success and continuity of the firm if their livelihood depends on the firm.

Empirically, Wen (2008) found that accounting conservatism is positively related to basic salary compensation and negatively related to equity-based compensation. Bushman et al. (2004) reported that executives' equity-based compensation varies with the timeliness of their earnings, while Yu (2021) found that salary is negatively associated with accounting conservatism.

On the basis of the previous discussion, the following is expected:

**Hypothesis 5a (H$_{5a}$).** *CEO salary is significantly and positively associated with the level of conservatism.*

**Hypothesis 5b (H$_{5b}$).** *CEO equity-based compensation is significantly and negatively associated with the level of conservatism.*

### 3. Data and Methodology

*3.1. Sample*

The main goal of this study is to examine how CEO characteristics are related to accounting conservatism in the Jordanian context. To achieve this objective, we constructed a sample from Jordanian industrial and service companies listed on the Amman Stock Exchange (ASE) during the period 2014–2021. Owing to the special features of financial companies, the financial sector is excluded from the sample. The final sample includes 672 yearly observations from both sectors. This study employs annual data, where the financial variables were extracted mainly from the website of ASE (www.ase.jo.com, accessed on 5 November 2022); however, the CEO characteristic variables were manually collected from the annual reports.

*3.2. Variable Measurements*

Dependent Variable

Accounting conservatism is the dependent variable in this study, calculated using the accrual-based measure developed by Givoly and Hayn (2000) and Givoly et al. (2007). The calculation of conservatism dimensionalised by accruals is income before interest and taxes, plus depreciation expenses, minus cash flow from operational activities; the sum of these two components is divided by the total assets at the beginning of the accounting period. As a result, the accrual value can then be dispersed across a practical number of years, possibly using an average with year *t* selected in the middle of the specified period. The timeframe selected spans 3 years. After that, the average is multiplied by $-1$, following Makhlouf et al. (2018) and considering that the most likely scenario is a reversal of accruals within 1

or 2 years (Ahmed and Duellman 2007). Accordingly, the following equations are used to calculate accounting conservatism:

$$Accruals_{it} = \beta_0 + \beta_1[(IBEI + DEP - CFO)]_{it}/TA_{t-1} + \varepsilon_{it} \tag{1}$$

$$AC_{it} = Accruals/(3\ years) - 1 \tag{2}$$

*IBEI* = income before interest and taxes.
*DEP* = depreciation expense for the current period.
*CFO* = cash flow resulting from operating activities.
$TA_{t-1}$ = lag value of total assets.

### 3.3. Independent Variables

The CEO characteristics represent the main independent variables in the study. These characteristics include CEO tenure, which is equal to the total number of years working as a CEO in the company. CEO experience is calculated as the total number of years that the CEO has served as a CEO during their life, even if in different companies; however, in case there is no experience in other companies, CEO experience equals CEO tenure. In addition, this study includes the level of education for the CEO: the value is set as one for their holding a bachelor's degree, while it is two for a master's degree, three for a PhD and four for others. The CEO's age is also included in our model, as well as their annual compensation. Two components represent CEO compensation: the first is salary calculated as the logarithm of total executive compensation, while the second measure is CEO ownership, measured as a percentage of stocks owned by the CEO at the end of the year. These variables are measured similarly to how prior studies in the literature have done so (Makhlouf et al. 2018; Alhmood et al. 2020; Degenhart et al. 2021).

### 3.4. Control Variables

Following the related studies, four control variables are included: firm size, measured as the natural logarithm of total assets; the M/B, which represents the market-to-book ratio; leverage, calculated as total debt divided by total assets; and the return on assets, calculated as net income divided by total assets. Table 1 offers all the variables' definitions and measurements.

**Table 1.** Summary of variables' definitions and measurement.

| Description | Variable Name | Measurement |
|---|---|---|
| Dependent variable: | | |
| Accounting Conservatism | AC | The accrual-based measure of conservatism |
| Independent variable: | | |
| CEO.AGE | Age | The CEO's age at the beginning of the year |
| CEO.TENURE | Ten | The number of years being served as a CEO |
| CEO.EDUCATION | EDU | The level of education: the value is set as one for a CEO's holding a bachelor's degree, two for a master's degree, three for a PhD and four for others. |
| CEO.EXPERIENCE | Exp | The total number of years that the CEO has served as a CEO during their life, even if in different companies |
| CEO.SALARY | SaL | The logarithm of total executive compensation |
| CEO.OWNERSHIP | OWN | The percentage of stocks owned by the CEO at the end of the year |
| Control variables: | | |
| SIZE | FZ | The total assets at the beginning of the year |
| Market-to-Book Ratio | MTB | The ratio of market value to the book value of equity |
| Leverage | LEV | The ratio of total debt to total equity |

### 3.5. Descriptive Statistics

Table 2 presents the descriptive statistics for our dependent, independent and control variables over the period 2015–2021, including mean, median, minimum and maximum observations and standard deviation. The results for the dependent variable, accounting

conservatism (AC), show that in the examined firms, the mean value recorded is 0.7%, with a standard deviation of 3.9%, which is consistent with average negative accruals because we have multiplied it by −1. These results are close to those reported by Yuliarti and Yanto (2017) and Ahmed et al. (2002). In addition, Table 2 shows that the average age of the CEOs in the study sample is approximately 53 years; the youngest CEO is 44 years old and the oldest is 62 years old. In the Jordanian context, this is in line with Bsoul et al.'s (2022) work, which found that the average age of CEOs is nearly 55 years.

**Table 2.** Descriptive statistics of research variables.

| Variable | Obs. | Mean | Std. Dev. | Min | Max |
|---|---|---|---|---|---|
| Accounting Conservatism | 672 | 0.007 | 0.039 | −0.045 | 0.061 |
| Age | 672 | 52.982 | 6.918 | 44.000 | 62.000 |
| Tenure | 672 | 4.797 | 2.732 | 2.000 | 9.000 |
| Education | 672 | 1.366 | 0.482 | 1.000 | 2.000 |
| Experience | 672 | 7.241 | 5.071 | 2.000 | 15.000 |
| Salary | 672 | 4.828 | 0.235 | 4.526 | 5.125 |
| Ownership | 672 | 0.050 | 0.119 | 0.001 | 0.411 |
| Firm Size | 672 | 7.410 | 0.332 | 6.993 | 7.835 |
| Market-to-Book Ratio | 672 | 0.807 | 0.343 | 0.410 | 1.306 |
| Leverage | 672 | 21.315 | 16.460 | 0.510 | 43.154 |

All the variables were defined in Table 1.

Table 2 shows that, on average, CEOs served for around 5 years in their positions (CEO tenure), where the longest period was 9 years. This result is consistent with the results reported by Huang et al. (2021), which indicate that CEOs served for around 5 years. As for the education level variable, this study shows that while CEOs in general possess postgraduate qualifications, this result is close to that found by Bsoul et al. (2022), who indicate the same. Regarding CEO experience, Table 2 shows that CEOs on average worked for 7 years in their positions, where the longest period was 15 years and the lowest period 2 years; this result is consistent with results found by Bsoul et al. (2022), who revealed that CEOs on average served for around 10 years.

In addition, Table 2 shows that when proxied by the natural logarithm of salary, the mean value for salary reaches 4.2. This result is close to that found by Bouaziz et al. (2020). However, our results indicate that CEOs have 5% ownership in their firms. In the Jordanian context, this result is a little higher than that reported by Qawasmeh and Azzam (2020), who reveal that average CEO ownership is about 3.4%. Finally, Table 2 provides statistics on the control variables in our study. The mean values reported for firm size, market-to-book ratio and leverage are around 7.4, 0.8 and 21, respectively.

*3.6. Method*

Following Yu (2021), this study adopts the following model to examine the relationship between CEO characteristics and accounting conservatism:

$$AC_{it} = \alpha_0 + \alpha_1 \text{Age}_{it} + \alpha_2 \text{Ten}_{it} + \alpha_3 \text{EDU}_{it} + \alpha_4 \text{Exp}_{it} + \alpha_5 \text{SaL}_{it} + \alpha_6 \text{OWN}_{it} + \alpha_7 \text{FZ}_{it} + \alpha_8 \text{M/B}_{it} + \alpha_9 \text{LEV}_{it} + \varepsilon_{it} \tag{3}$$

where $AC_{it}$ represents accounting conservatism, Age represents CEO age, Ten represents CEO tenure, EDU represents CEO level of education, Exp represents CEO experience, SaL represents CEO salary, OWN represents CEO ownership, FZ represents company size, M/B represents market-to-book ratio and LEV represents leverage.

The regression analysis is carried out stepwise and sequentially for a better representation of the relationships between the dependent and independent variables. Table 3 summarises and presents the results of the tests that have been conducted to test for endogeneity, heteroscedasticity, cross-sectional dependency and serial correlation to ensure that we employed the most suitable statistical model to test our hypotheses and to obtain

reliable estimates of the parameters of the model. As presented in Table 3, the first step aims to discover whether there is causality between AC and CEO variables and vice versa. Thus, the Durban–Wu–Hausman test for endogeneity was employed. The output of this test reveals that the possibility of exogeneity (the possibility of influences by unchosen independent variables) should be accepted. Next, this study conducts the homogeneity Fisher test to select either OLS or panel analysis. The output of the Fisher test reveals that it generates values (statistics) higher than the critical values; this is taken to indicate that there are specific effects, and accordingly, panel analysis is more suitable. In addition, the Hausman test, which is the best alternative for checking differences between random and nonrandom (fixed) effects, is used. The test result, as shown in Table 3, indicates that the fixed effects model is more appropriate.

**Table 3.** Results of the analysis selection tests.

| Test | Stat. | Prob. |
| --- | --- | --- |
| Homogeneity Fisher test | 7.488 | 0.0000 |
| Specification Hausman test | 92.324 | 0.0000 |
| Heteroscedasticity Breusch–Pagan test | 125.07 | 0.0210 |
| Cross-sectional dependence Pesaran test | 132.967 | 0.0000 |
| Serial correlation Wooldridge test | 14.828 | 0.0000 |
| Endogeneity Durbin–Wu–Hausman test | 1.729 | 0.1635 |

In addition, the Breusch–Pagan test is carried out, which confirms the presence of heteroscedasticity. The heteroscedasticity test is used because it shows, on the basis of using regression output, whether errors of variances are due to the reliance of dependent variables on independent variables. The results of the test indicate that heteroscedasticity is significant at 5%. This is also confirmed through chi-square statistics thanks to the existence of heteroscedasticity in the residuals. Thus, the OLS model is not used.

The significant level of the cross-sectional dependency of the models chosen is revealed through the Pesaran test. In addition, the Wooldridge test is used to check whether heteroscedasticity and serial correlation exist. In such an eventuality, the feasible generalised least squares (FGLS) technique would be used.

## 4. Results and Discussion

### 4.1. Multicollinearity Check

Table 4 depicts the correlation matrix coefficients by using Pearson's correlation, which was employed to test for the existence of multicollinearity. Multicollinearity exists when independent variables are correlated to each other, which in turn undermines the statistical significance of the estimated coefficients. It is suggested that the correlation should not exceed 0.8, to prove that there is no multicollinearity problem between the variables (Gujarati 2004). As shown in Table 4, the highest correlation is between CEO ownership and CEO salary, with an amount of 0.562; this indicates that there is no multicollinearity problem between the independent variables. Furthermore, Table 3 presents the variance inflation factors (VIFs) test, which was used to detect the multicollinearity degree between the variables. The VIF for all variables ranged from 1.04 to 2.29, which is below 10 (Greene 2008). Thus, the possibility of a multicollinearity problem was not present in the analysis.

**Table 4.** The results of the tests of multicollinearity.

| Variables | Age | Tenure | Education | Experience | Salary | Ownership | Firm Size | MB | Leverage | VIF |
|---|---|---|---|---|---|---|---|---|---|---|
| (1) Age | 1.000 | | | | | | | | | 1.24 |
| (2) Tenure | 0.085 | 1.000 | | | | | | | | 2.29 |
| (3) Education | −0.129 | −0.168 | 1.000 | | | | | | | 1.04 |
| (4) Experience | 0.008 | −0.136 | 0.136 | 1.000 | | | | | | 2.22 |
| (5) Salary | −0.045 | 0.079 | 0.047 | 0.023 | 1.000 | | | | | 1.26 |
| (6) Ownership | 0.107 | 0.419 | 0.471 | 0.516 | 0.562 | 1.000 | | | | 1.07 |
| (7) Firm Size | 0.027 | 0.085 | 0.168 | −0.096 | 0.127 | 0.143 | 1.000 | | | 1.20 |
| (8) MB | 0.019 | −0.053 | 0.029 | −0.028 | −0.096 | −0.073 | −0.014 | 1.000 | | 1.10 |
| (9) Leverage | −0.163 | 0.115 | 0.067 | 0.005 | 0.340 | 0.239 | 0.344 | −0.238 | 1.000 | 1.11 |

All the variables were defined in Table 1.

### 4.2. Unit Root Test

Table 5 presents the stationarity of time series which is tested using the Levin et al. (2002) test (LLC), where the null hypothesis assumes that panel data has unit roots. Before proceeding to the main analysis, the durability of the variables is to be analysed to ensure that all variables are stationary i.e., the mean, variance and covariance of the variable are fixed during the time.

**Table 5.** The results of stationary tests.

| Variables | LLC-Statistic | *p*-Value | Result |
|---|---|---|---|
| Accounting Conservatism | −30.3140 | 0.0000 | Stationary at Level |
| Age | −14.1399 | 0.0000 | Stationary at Level |
| Tenure | −13.9020 | 0.0000 | Stationary at Level |
| Education | −12.8422 | 0.0000 | Stationary at Level |
| Experience | −7.6020 | 0.0000 | Stationary at Level |
| Salary | −12.0688 | 0.0000 | Stationary at Level |
| Ownership | −11.4820 | 0.0000 | Stationary at Level |
| Firm Size | −6.8315 | 0.0000 | Stationary at Level |
| Market-to-Book Ratio | −13.9456 | 0.0000 | Stationary at Level |
| Leverage | −112.3732 | 0.0000 | Stationary at Level |

The results of this test are presented in Table 5 and confirm that all the variables included in the analysis are stationary at level and statistically significant, based on *p*-values in the table below, leading us to accept the alternative hypothesis.

Table 6 presents the main results of this study using FGLS analysis. Regarding the CEO age, as shown in Table 6, the coefficient of CEO age is positive yet insignificant. This indicates that accounting conservatism in Jordanian companies is not affected by CEO age, which is consistent with the findings of Adyunita et al. (2021), using Indonesian data, and Makhlouf et al. (2018) from Jordan. However, it is inconsistent with upper-echelon theory and the common notion that older managers tend to be more conservative.

**Table 6.** Estimation results of the panel FGLS model.

| Accounting Conservatism | Coef. | Z-Value | *p*-Value |
|---|---|---|---|
| Age | 0.0001 | 0.98 | 0.326 |
| Tenure | 0.0021 | 2.74 | 0.006 *** |
| Education | 0.0001 | 1.21 | 0.218 |
| Experience | 0.0015 | 3.01 | 0.003 *** |
| Salary | 0.0073 | 1.34 | 0.179 |
| Ownership | 0.0001 | 0.88 | 0.471 |
| Firm Size | 0.0006 | 0.19 | 0.847 |
| Market-to-Book Ratio | 0.0162 | 5.25 | 0.000 *** |
| Leverage | 0.0002 | 2.70 | 0.007 *** |
| Constant | −0.0622 | −1.94 | 0.053 * |
| Number of Groups | 96 | Number of Time Periods | 7 |
| R | 0.5287 | R$^2$ | 0.2795 |
| Number of Obs. | 672 | Wald Chi-Square | 74.37 (0.0000) |

\* Significant at 10% level. \*\*\* significant at 1% level.

An explanation for this result is that the relation between risk taking and age depends on the context. Therefore, risk taking behaviour changes with different types of decision rather than with different age groups; for example, as Mata et al. (2011) reported, risk taking depends on whether the risks associated with the decision are known beforehand. In the same vein, Eberhardt et al. (2019) found that age is not related to the ability or motivation to think about and manipulate numerical information. Accordingly, based on our results, CEO age is not a determinant of financial reporting quality. Consequently, the first hypothesis which states that *CEO age is significantly and positively associated with the level of conservatism* is rejected.

As shown in Table 6, CEO tenure is positively and significantly related to accounting conservatism. This reveals that long-serving CEOs are more conservative. Our finding is in line with that found by Dong and Gai (2017), who report a positive association between CFO tenure and the percentage of asset impairment provision. Several explanations can be found for this result. First, managers fearing the loss of their built reputation and aspiring to keep their position tend to play it safe and avoid aggressive accounting practices. Second, and consistent with the tenets of upper-echelon theory, the longer a CEO stays in a firm, the more likely their perspective and knowledge base will become limited and their options and strategies will be less innovative (Hambrick and Mason 1984). Third, long-serving CEOs are under less pressure to prove their abilities to the market compared with newly appointed CEOs. Therefore, the second hypothesis, which proposes that *CEO tenure is significantly and positively associated with the level of conservatism*, is accepted.

CEO education seems to have no effect on accounting conservatism; the coefficient on education is positive but insignificant. Our finding is consistent with the argument of Pfeffer (1981), who contends that education in management serves only as a filtering device for staffing and enrolling individuals in jobs; therefore, such education will not have a significant effect on the individual or the firm. This is inconsistent with the positive relationship between directors' education and conservatism found by Makhlouf et al. (2018), in Jordan. According to our results, specialised knowledge and technical skills obtained through education do not affect CEOs' attitudes towards conservatism; rather, skills obtained through real-world experiences do, as shown by the next finding. Accordingly, the third hypothesis, which states that *CEO education is significantly and positively associated with the level of conservatism*, is rejected.

Table 6 also indicates that CEOs with higher positive and significant experiences are related to accounting conservatism. This positive result is expected; according to previous studies, CEOs with more experience and skills are expected to be more conservative, signalling their competency to the market. Additionally, more-experienced managers care more about their reputation and have more to lose in the case of firm failure; thus, they tend to exercise higher levels of conservatism (Baik et al. 2011). This is in line with findings by Yunos et al. (2014) and Haider et al. (2021), who found that managers with more experience typically report in a more conservative manner. However, the result is inconsistent with the expectations of upper-echelon theory, which states that experience outside the firm increased their tendency to take risks. Thus, our fourth hypothesis, which proposes that *CEO experience is significantly and positively associated with the level of conservatism*, is accepted.

CEO compensation, under both components, i.e., salary and ownership, is found to be positively but insignificantly related to accounting conservatism in Jordanian companies. This indicates that increasing CEOs' salaries or granting them more equity does not affect their accounting reporting behaviour. These results are similar to those reported by Aburisheh et al. (2022), who indicated an insignificant relation between managerial ownership and conservatism in Jordanian firms. Thus, both hypotheses regarding CEO compensation are rejected. The result suggests that payment, regardless of its type, has no impact on Jordanian managers' conservative accounting practices.

Regarding control variables, firm size has no effect on conservatism, while both market-to-book ratio and leverage have positive and significant effects on accounting conservatism.

These findings are consistent with previous Jordanian studies, such as those obtained by Makhlouf et al. (2018) and Alhmood et al. (2020), among others.

## 5. Conclusions

This study examines the relationship between CEOs' characteristics (age, tenure, experience, education and compensation) and the accounting conservatism calculated by the accrual-based measure for a sample of 672 yearly observations from both industrial and service companies in Jordan that are listed on the Amman Stock Exchange (ASE) for the period 2014–2021.

The findings provide evidence that CEO tenure and experience positively and significantly contribute to accounting conservatism. Additionally, market-to-book ratio and leverage are positively and significantly associated with levels of conservatism. However, no significant relationship is found between CEO education, age and compensation on one hand and accounting conservatism on the other.

Consistent with upper-echelon theory, the results indicate that an increase in both CEO experience and CEO tenure tends to prevent aggressive accounting practices and improve financial reporting by increasing conservative accounting practices.

This study contributes to the pertinent literature by providing evidence of the importance of recognising the effects of CEOs' characteristics on influencing accounting conservatism. Implications of this study are as follows: First, it is recommended, on the basis of these results, that researchers take CEO characteristics into consideration when analysing the determinants of accounting conservatism because they affect the quality and practice of financial reporting. Second, policymakers in Jordan are recommended to consider the attributes of the CEOs when revising governance codes because they currently focus only on the attributes of the board of directors. Moreover, the current research provides social implications for investors who suffer from information asymmetry; investors are recommended to analyse the attributes of managers before making an investment decision as manager attributes affect the quality of financial reporting.

Finally, future studies should explore the relationship between accounting conservatism and different CEO characteristics.

**Author Contributions:** Conceptualisation, R.A.; methodology, R.A. and B.K.; software, R.A. and S.A.; validation, R.A. and B.K.; formal analysis, R.A. and S.A.; investigation, R.A. and S.A.; resources, R.A.; data curation, R.A. and R.H.; writing—original draft preparation, S.A. and R.H.; writing—review and editing, R.A.; visualisation, B.K.; supervision, R.A. All authors have read and agreed to the published version of the manuscript.

**Funding:** This research received no external funding.

**Informed Consent Statement:** Not applicable.

**Data Availability Statement:** The data that support the findings of this study are available from the Amman Stock Exchange's official website: https://www.ase.com.jo/en/disclosures (accessed on 5 November 2022).

**Conflicts of Interest:** The authors declare no conflict of interest.

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
