# Peer review of "Do CEO Attributes Spur Conservatism?"

_ijfs, doi:10.3390/ijfs11010052_

Round 1

Reviewer 1 Report

Comments and Suggestions for Authors

I review the paper as it has potential but it is looking for me an early draft. In the introduction paragraph of your review, you should clearly and succinctly summarize the paper's main contribution. This should include a brief description of the research question or problem, the methods used to address it, and the main findings or conclusions. It is also important to comment on the originality and significance of the contribution, highlighting any gaps in the existing literature that the paper addresses.

In the conclusion of your review, you should discuss the practical and social impact of the paper. This should include a discussion of how the findings or recommendations presented in the paper could be used to improve practice or policy in the relevant field. You should also consider any broader social or ethical implications of the research, such as its impact on marginalized communities or on issues of equity and justice.

When analyzing the tables in the paper, it is important to provide a clear and detailed explanation of each table, highlighting the key results and what they contribute to the study's overall findings. You should also comment on the clarity and usefulness of the table, making suggestions for how it could be improved if necessary.

Finally, when discussing the literature, you should ensure you reference up-to-date and relevant sources. This will help to demonstrate your expertise in the field and your understanding of the current state of research on the topic. You should also critically evaluate the literature, highlighting any gaps or limitations in the existing research and explaining how the paper you are reviewing contributes to filling those gaps.

Author Response

Reviewer 1 comments and author’s responses 1. I review the paper as it has potential, but it is looking for me an early draft. In the introduction paragraph of your review, you should clearly and succinctly summarize the paper's main contribution. This should include a brief description of the research question or problem, the methods used to address it, and the main findings or conclusions. It is also important to comment on the originality and significance of the contribution, highlighting any gaps in the existing literature that the paper addresses. Response: We would like to thank the reviewer for this comment. Accordingly, the researchers have acted upon it and expanded the introduction to include the above-mentioned points. Please refer to the lines 44-50 and the lines 57-63. 2. In the conclusion of your review, you should discuss the practical and social impact of the paper. This should include a discussion of how the findings or recommendations presented in the paper could be used to improve practice or policy in the relevant field. You should also consider any broader social or ethical implications of the research, such as its impact on marginalized communities or on issues of equity and justice. Response: We would like to thank the reviewer for this comment. Accordingly, the researchers have acted upon it and revised the conclusion. Please refer to the lines 490- 500 3. When analyzing the tables in the paper, it is important to provide a clear and detailed explanation of each table, highlighting the key results and what they contribute to the study's overall findings. You should also comment on the clarity and usefulness of the table, making suggestions for how it could be improved if necessary. Response: We would like to thank the reviewer for this valuable comment. Acting upon this comment the authors have changed the titles of the tables and provided an explanation preceding each table as requested. 4. Finally, when discussing the literature, you should ensure you reference up-to-date and relevant sources. This will help to demonstrate your expertise in the field and your understanding of the current state of research on the topic. You should also critically evaluate the literature, highlighting any gaps or limitations in the existing research and explaining how the paper you are reviewing contributes to filling those gaps. Response: We would like to thank the reviewer for this comment. Accordingly, the researchers have acted upon it and updated the literature section of this research to include more recent research on this area of investigation. Please refer to the lines 110-112, line 121 and the lines 131-145.

Reviewer 2 Report

Comments and Suggestions for Authors

Dear authors,

The topic of the study is interesting, but according to the following conditions, the paper needs major revision

1- Why has the study been concentrated on Jordan?

2-What is the research problem and motivation of the study?

3- The research implications should be highlighted in the study

4- The theoretical issues and literature sections should be up to date with the recent studies; some are suggested as follows:

1-How Do Employees React When Their CEO Speaks Out? Intra- and Extra-Firm Implications of CEO Sociopolitical Activism

2-The effect of cash flow information asymmetry criteria on conservatism in Iran

3-The relationship between board characteristics and social responsibility with firm innovation

4-CEO Facial masculinity and accounting conservatism

5-The relationship between managerial entrenchment and accounting conservatism

Author Response

Reviewer 2 comments and author’s responses

  1. Why has the study been concentrated on Jordan?

Response: We would like to thank the reviewer for this valuable comment which helped enhancing the introduction section indeed. Acting upon this comment, the researchers have added a paragraph in the introduction section. Please refer to the lines 65-75.

  1. What is the research problem and motivation of the study?

Response: Thank you for this valuable comment, acting upon it the researchers have revised the introduction section to emphasize the problem and the motivation of the study. Please see the lines 44-50 and the lines 57-63.  

  1. The research implications should be highlighted in the study

Response: we would like to thank the reviewer. Implications have been added to the conclusion section, lines 490-500.

  1. The theoretical issues and literature sections should be up to date with the recent studies; some are suggested as follows:

1-How Do Employees React When Their CEO Speaks Out? Intra- and Extra-Firm Implications of CEO Sociopolitical Activism

2-The effect of cash flow information asymmetry criteria on conservatism in Iran

3-The relationship between board characteristics and social responsibility with firm innovation

4-CEO Facial masculinity and accounting conservatism

5-The relationship between managerial entrenchment and accounting conservatism

Response: We thank the reviewer for this valuable comment, the literature review part of this research has been updated to include more recent research and the mentioned papers have been referred to in the literature section. Please refer to the lines 110-112, line 121 and the lines 131-145.

Round 2

Reviewer 1 Report

Comments and Suggestions for Authors

Thank you for addressing my comments. I suggest accepting the paper for publication. 

Reviewer 2 Report

Comments and Suggestions for Authors

Dear authors,

Thank you very much for sending your revised paper. The current version meets my academic expectations.